# A Hybrid Method for Predicting Traffic Congestion during Peak Hours in the Subway System of Shenzhen

**DOI:** 10.3390/s20010150

**Published:** 2019-12-25

**Authors:** Zhenwei Luo, Yu Zhang, Lin Li, Biao He, Chengming Li, Haihong Zhu, Wei Wang, Shen Ying, Yuliang Xi

**Affiliations:** 1School of Resources and Environmental Science, Wuhan University, Wuhan 430079, China; luowei1993@whu.edu.cn (Z.L.);; 2RE-Institute of Smart Perception and Intelligent Computing, Wuhan University, Wuhan 430079, China; 3School of Architecture and Urban Planning, Shenzhen University, Shenzhen 518000, China; 4Chinese Academy of Surveying and Mapping, 28 Lianghuachi West Road, Haidian Qu, Beijing 100830, China

**Keywords:** subway, congestion, traffic model, origin-destination (OD) matrix, agent-based simulation

## Abstract

Traffic congestion, especially during peak hours, has become a challenge for transportation systems in many metropolitan areas, and such congestion causes delays and negative effects for passengers. Many studies have examined the prediction of congestion; however, these studies focus mainly on road traffic, and subway transit, which is the main form of transportation in densely populated cities, such as Tokyo, Paris, and Beijing and Shenzhen in China, has seldom been examined. This study takes Shenzhen as a case study for predicting congestion in a subway system during peak hours and proposes a hybrid method that combines a static traffic assignment model with an agent-based dynamic traffic simulation model to estimate recurrent congestion in this subway system. The homes and work places of the residents in this city are collected and taken to represent the traffic demand for the subway system of Shenzhen. An origin-destination (OD) matrix derived from the data is used as an input in this method of predicting traffic, and the traffic congestion is presented in simulations. To evaluate the predictions, data on the congestion condition of subway segments that are released daily by the Shenzhen metro operation microblog are used as a reference, and a comparative analysis indicates the appropriateness of the proposed method. This study could be taken as an example for similar studies that model subway traffic in other cities.

## 1. Introduction

Urban traffic congestion has produced enormous challenges for both the daily lives of people and the stable development of societies and economies [1]. The alleviation of traffic congestion is essential and urgent. One solution involves predicting traffic congestion [2,3], which is a vital component of intelligent transportation systems (ITSs). Existing studies of traffic congestion prediction focus primarily on road traffic [4,5], and few studies have examined urban rail transit (URT) systems. Subways, which represent the main form of URT systems [6], currently play an increasingly significant role in traffic systems and have become a major mode of transportation in many metropolitan areas [7]. Subway congestion, which usually refers to the crowding of people in carriages, causes reductions in comfort [8] and increases travel time when the passengers cannot board crowded subway carriages. Such subway congestion increases delays, and the onset of an emergency situation under congestion could adversely affect evacuation [9].

The forecasting of traffic flow has always been an important research content of sensor networks. Previous studies on the forecasting of traffic flow and the prediction of congestion have used various methods. These methods can be divided into several categories, specifically linear, nonlinear, and traffic simulation models. Early studies employed primarily linear models, such as nearest neighbor (NN) [10] and historical average (HA) models [11]. NN and HA models are simple and easily applied, but they do not reflect the time-varying properties of traffic flow. Hence, time series models, such as autoregressive integrated moving average (ARIMA) models [12] and seasonal autoregressive integrated moving average (SARIMA) models [13], have been proposed and display better performance. However, time series models require continuous inputs of substantial amounts of data [14,15], and such models do not account for information from adjacent segments. Kalman filtering (KF) models [16], which require limited streams of data, were first employed in traffic flow forecasting by Okutani and Stephanedes [17]. KF models can be easily implemented on a computer and have high precision and stability; however, the parameters of such models must be recalculated each time a prediction is made, which leads to substantial computing requirements. All of the linear models have limitations, as traffic systems display considerable randomness, uncertainty and nonlinearity, and these models become less accurate as the prediction interval decreases in length.

Nonlinear models are data-driven [18]; that is, these models learn and generalize the movement patterns of traffic flows from massive amounts of observational data. Nonlinear models are more suitable than linear models in simulating traffic flows, given the uncertainty and nonlinearity of such flows. The nonlinear models commonly used in traffic flow analysis and congestion prediction include nonparametric regression (NPR) [19,20], neural network [21,22,23,24] and support vector machine (SVM) models [25,26]. NPR models search a collection of historical observations for records similar to the current conditions and use these records to estimate future system states [27]; such models are suited for use in traffic flow forecasting in the presence of special events. However, NPR models have several shortcomings, mainly that they require huge historical databases and the complexity of searching such databases for similar states. Traffic assignment model [28] is a common method for traffic departments to deal with traffic congestion. It is based on the assumption that road users’ behavior can be predicted by user equilibrium equations, and therefore incorporating more realistic behavior principles suffer from solution methodology convergence issues [29], limiting practical application. Neural network models display strong self-adaptive learning ability and anti-interference properties, but such models cannot minimize the expected risk, require ample amounts of time to train using sample data, and the trained models are applicable only to the current segments. SVM models that use the principle of structural risk minimization address some of the inherent defects of neural networks [30] and display improved robustness and generalization ability [31]. However, the algorithms used to train SVM models are slow and computationally complex and have large memory costs.

In fact, a growing number of studies have combined various methods to make use of their respective advantages [20,32,33]. These hybrid models produce predictions with improved accuracy, but the forecasting speed of these models is reduced and they are more complex than the individual models. Although the above-mentioned methods have achieved promising outcomes, they are all mathematical modeling methods, and they cannot represent the dynamic processes of traffic systems. Traffic simulations that concentrate on traffic entities and their interactions provide an effective means of representing traffic-related processes and analyzing the temporal-spatial characteristics of traffic flow. Cetin and Burri [34] demonstrated that computer simulations of traffic systems provide an effective and risk-free platform to investigate the nature of transportation systems from different perspectives. Traffic simulation models have been extensively applied to traffic flow forecasting in the past [35,36,37]. In addition, agent-based modeling (ABM), which offers an intuitive way to describe every autonomous entity on the individual level [38], has drawn increasing attention. Many studies have predicted traffic flow with agent-based simulations [39,40,41]. However, as described by Lu et al. [30], traffic simulation methods require real traffic flow data to predict traffic flow and congestion. In previous work, which mainly addresses road traffic, traffic congestion is usually predicted using GPS traces [42] or floating car data [43]. However, Sun et al. [32] argued that passenger flows in subway systems are characterized by regularity and randomness; thus, they differ from the traffic flows associated with other forms of public transportation. The above studies indicate that simulations provide an effective means of modeling traffic flow, and simulation models can be successfully built but depend strongly on the application and available data sources. Simulation models are rarely applied to subway traffic congestion prediction, due to the limited availability of real subway traffic data.

To address this problem, a hybrid method that combines a static traffic assignment model with an agent-based dynamic traffic simulation model is proposed in this study to predict recurrent congestion in a subway system during peak hours. In this study, Shenzhen, China, where the subway system has become the major mode of transportation, is taken as a case study. The residence and workplace data in this city are collected and taken as the OD matrix, with which the information on the trips of the residents can be generated by calling Baidu Map Services (BMS). An agent-based framework (Figure 1) is developed to simulate the travel process using those trips. The passenger flow size distribution is extracted from the simulation results, and the underlying congested segments in the subway system can be detected. The remainder of this paper is organized as follows. Section 2 introduces the basic principles and methodology. Next, a case study for the subway system of Shenzhen is presented in Section 3. Section 4 demonstrates the results and analyzes the test scenario. Section 5 then concludes this paper and discusses future research issues.

## 2. Methodology

### 2.1. A Static Traffic Assignment Model

Static traffic assignment methods distribute traffic demand to a network with the preconditions of fixed origins and destinations. This study focuses on commuting behavior during the morning peak hours [44,45,46] on workdays. Hence, the homes and workplaces of residents are acquired from the building property registration information and social insurance records collected by the government and are utilized to construct an OD matrix that reflects the traffic demand. Commuting passengers occupy a large proportion of the flow volume of subway transit during this period. With this OD matrix, traffic demand is assigned statically to the subway network by generating the trips beforehand.

A trip, which consists of a departure time, a travel mode, the related stations and a transfer plan, determines the travel behavior of travelers. Some studies have concentrated on the acquisition of the trips of travelers. For example, Me et al. [47] extracted trip information from transit smart card data and used it to identity the commuters accurately; however, these trips lack the portions during which travelers move from their homes to the originating stations and from the destination stations to their workplaces. Huang et al. [48] effectively studied trip chaining behavior using real-time GPS travel data. However, personal privacy cannot be neglected when research is conducted using data from thousands or even millions of citizens. To address this problem, BMS is selected in this study to generate unbroken trips with efficiency and safety.

The Baidu Map development platform (BMDP) provides access to a series of BMS that can be used by registered map developers. The two map services that are used to generate trips with an OD matrix are Geocoding and Routing and Transportation. The original OD matrix stores the homes and workplaces of citizens in the form of detailed addresses. The encoding service transforms these addresses into coordinates in a specific geographic coordinate system, usually WGS84. Next, taking the coordinate pairs as inputs, the Routing and Transportation service searches for optimal solutions that enable a traveler to reach his or her destination. A trip can be created with this solution that includes the necessary travel information. In consideration of the popularization of map apps and the convenience of the travel solutions they provide, the generated trips have high consistency with reality.

Subway traffic congestion is a phenomenon in which passenger flow volumes exceed the capacity of subway infrastructure over short periods, and time is a crucial parameter that cannot be reflected by static traffic assignments. Hence, a dynamic traffic simulation model that represents the temporal and spatial variations in subway passenger flow on the basis of the generated trips, is introduced in the next section.

### 2.2. An Agent-Based Dynamic Traffic Simulation Framework

As described by Ferber [49], an agent is a physical or virtual entity that can act, perceive its environment and communicate with others; is capable of autonomous action; and has skills to achieve its goals and tendencies. Agent-based simulation is a powerful tool for handling problems involving complex systems, because some perspectives suggest that complex systems emerge from the bottom up, are highly decentralized and are composed of a multitude of heterogeneous objects called agents [50]. Agent-based traffic simulations provide an effective means of representing the dynamics and interactions of large numbers of traffic entities, such as passengers, vehicles and the subway network and reflect the dynamic variations in the temporal and spatial characteristics of passenger flows, from which subway traffic characteristics can be extracted.

In this study, an agent-based traffic simulation framework is designed and implemented to explore the distribution characteristics of subway passenger flows during peak hours to predict recurrent congestion. In this framework, “simulation environment” refers to the subway network, and traffic entities are modeled as agents; thus, a multi-agent system (MAS) can be established. The framework takes the trips generated above as its inputs and it has a four-tier architecture, as shown in Figure 1.

#### 2.2.1. Action-Driven System

The constructed MAS includes three types of agents, person agents, station agents and vehicle agents, and is an action-driven system. The actions for subway vehicle agents include (1) entering a subway station and (2) departing from a subway station. In addition, the actions for person agents are (1) setting out, (2) arriving at a subway station, (3) leaving a subway station, (4) boarding the subway, (5) disembarking from the subway, and (6) arriving. An event comprises an action and the related time attributes.

The system is motivated by a global event queue, which stores series of events in chronological order. Many events are generated during each simulation, and each event has the “start time” and “finish time” attributes. When an event occurs, its finish time will be calculated, and the start time of the next event for the associated agent will be assigned. For example, when a person sets out, a “person set out event” is created, and its start time can be calculated according to the empirical travel time. This event will be performed when the system time reaches its start time, and the next event for this person could be “person arrive at station event” is created and inserted into the event queue in chronological order. The system continues to run through fetching events from the event queue to perform and inserting newly generated events (if they exist) until the stopping conditions that all of the travelers reach their destinations or the elapsed time reaches a specified value are satisfied.

#### 2.2.2. Simulation Workflow

The simulation workflow used in the proposed framework is presented in Figure 2. Initial demand prepares for the simulation, including constructing the environment and creating the agents. The use of mobile simulation (Mobsim) means that all of the agents execute their actions based on designated rules and their perceptions of the environment until all of the travelers reach their destinations. Scoring evaluates the performance of a trip based on the elapsed time and travel distance. When a traveler is late, the score of the corresponding trip is far lower than those of other trips. All of the late travelers need to adjust their trips by setting out earlier in the next iteration (i.e., replanning). A relaxed state is defined as the first time when none of the travelers are late. The relevant information can then be output to permit analysis of the passenger flow characteristics.

With the proposed framework and the generated trips, it is possible to simulate the whole traffic process, and the distribution patterns of the passenger flow can be extracted. In addition, the congested segments in a subway system can be detected by examining the passenger flow distribution and calculating the values of specific traffic indexes.

### 2.3. Passenger Flow Distribution

The features of passenger flow distributions within specified periods can reveal subway traffic congestion directly and effectively. In Figure 3, a sample subway network with 8 nodes and 7 edges is shown, and the corresponding passenger flow network has been constructed to clarify the statistical traffic indexes.

In Figure 3a, every node represents a subway station and the intersections (Node B) of edges represent transfer stations. A subway segment (or section) refers to an edge between two stations. Taken together, the stations and segments constitute the subway network. Additionally, the directed edges in Figure 3b distinguish the upward and downward flows of passengers.

In the following descriptions, i is the ordinal enumeration of the stations and segments. For example, segment 1 in Figure 3a is segment AB for Line 1 in the direction from A to D. The related passenger flow parameters are defined as follows:

${\mathrm{UF}}_{i}^{\mathrm{on}}$ indicates the onboard flow of station i in the upward direction; ${\mathrm{UF}}_{i}^{\mathrm{off}}$ indicates the alighting flow of station i in the upward direction; ${\mathrm{DF}}_{i}^{\mathrm{on}}$ indicates the onboard flow of station i in the downward direction; and ${\mathrm{DF}}_{i}^{\mathrm{off}}$ indicates the alighting flow of station i in the downward direction;

Segment passenger flow (SPF) is defined as the passenger flow volume of a segment. ${\mathrm{USPF}}_{i}$ represents the SPF of segment i in the upward direction, whereas ${\mathrm{DSPF}}_{i}$ represents the SPF of segment i in the downward direction. Then,
(1)$${\mathrm{USPF}}_{i}=F_{i}^{\mathrm{on}}-F_{i}^{\mathrm{off}},\;i=1$$
(2)$${\mathrm{USPF}}_{i}={\mathrm{USPF}}_{i-1}+F_{i}^{\mathrm{on}}-F_{i}^{\mathrm{off}},\;i>1$$
and
(3)$${\mathrm{USPF}}_{i}={\mathrm{USPF}}_{i-1}+F_{i}^{\mathrm{on}}-F_{i}^{\mathrm{off}},\;i>1$$
(4)$${\mathrm{USPF}}_{i}={\mathrm{USPF}}_{i-1}+F_{i}^{\mathrm{on}}-F_{i}^{\mathrm{off}},\;i>1$$

Formulas (1)–(4) can be used to calculate the SPF of subway lines for the whole travel process when all of the passengers have reached their destination.

Many factors have been used to reflect the congestion situation of public transportation. Examples include vehicle speed and the ratio of distance and time. In view of the features of subway transit, the load factor of subway vehicles is used in this method to reflect the congestion state of the segment. β denotes the load factor of subway vehicles, where
(5)$$\beta =\frac{{\mathrm{SPF}}_{\max,i,t}}{C_{p}\;\times {\;N}_{t}\;\times \;\mu },$$

${\mathrm{SPF}}_{\max,i,t}$ represents the max SPF in segment i during time span t; $C_{p}$ represents the designed passenger capacity of the subway vehicles; $N_{t}$ indicates the number of trains that depart during time t; and μ is the adjustment factor, where
(6)$$\mu =\frac{C_{s}}{C_{r}},$$

$C_{s}$ is the capacity of travelers examined in this study, and $C_{r}$ is the actual number of daily subway passenger-trips made during the morning peak period. A classification diagram can be produced based on the value of β to demonstrate the three congestion states, smooth, crowded and congested.

## 3. Case Study

### 3.1. Study Area

#### 3.1.1. Shenzhen City

Shenzhen is used as an example in this study. Shenzhen [51] is a coastal metropolis that is located in the southern part of China and adjacent to Hong Kong; it lies between the longitudes of 113°46′ and 114°37′ east and between the latitudes of 22°27′ and 22°52′ north. The total area of Shenzhen is approximately 1952.84 km^2^, and the population of Shenzhen is nearly 10.78 million, meaning that Shenzhen is among the most densely populated cities. Shenzhen is the first special economic zone in China and is regarded as a window on the reform and opening up; thus, it occupies a significant economic position. In addition, Shenzhen contains 10 districts that are shown in Figure 4 [52]. In addition, buses, subways and taxis are the main modes of public transportation in Shenzhen, and Table 1 records the passenger flow volume and annual changes for these three modes of transportation [45]. The subway system displays a large increase in passenger flow volume in the public transportation system of Shenzhen.

#### 3.1.2. Subway Network

Shenzhen opened its first subway line in December 2004 and became the sixth city in China with a subway system. The system is operated by Shenzhen Metro Corporation (SZMC) and MTR <Shenzhen>. The subway system of Shenzhen presently includes eight subway lines (Nos. 1, 2, 3, 4, 5, 7, 9, and 11) and 168 stations, and the total length of the subway lines is 285 km. The subway network covers six districts, including Luohu, Futian, Nanshan, Bao’an, Longhua, and Longgang. In 2017, the daily passenger flow volume of the URT system in Shenzhen was 4.53 million person-trips and grew at a rate of 27.54%. The maximum daily passenger flow volume for the subway system of Shenzhen is 5.61 million trips. The total number of planned subway lines is 32, and the total length of these lines will be 1142 km by 2030. Undoubtedly, the subway system is receiving additional urban traffic volume in terms of both quantity and proportion, which increases the need to predict traffic congestion in this system. Table 2 provides basic information on the existing subway lines. The spatial distribution of the subway lines within the city of Shenzhen can be found in Figure 4, and Figure 5 shows this structure in greater detail, as well as the locations of the stations. Data provided by Shenzhen Research Center of Digital City Engineering.

#### 3.1.3. Subway Commuters

Commuting is a behavior in which people travel between their homes and workplaces and back again, and the term “subway commuters” means those who commute by subway. Because this study does not consider multimodal traffic, the target travelers exclude people who transfer between the subway and other modes of public transportation. Considering the service radius of subway stations, people whose homes and workplaces are both located within 1000 m of a station were selected for examination in this experiment. A spatial buffer analysis [53] based on real road network was conducted to identify applicable experiment objects, and 833,760 individuals were selected. The distributions of the homes and workplaces of the selected citizens are shown in Figure 6 and Figure 7. These homes and workplaces constitute the OD matrix for the travels of those commuters during the morning peak hours on workdays.

### 3.2. Detailed Trips

Detailed travel plans made up of walking distances, subway routes and directions, stations, transfer strategies and empirically determined travel times are needed to guide travelers to their destinations. The trip generation method is described in Section 2.1, above. In fact, several travel plans were recommended for a traveler by BMS. The optimal one was selected, and those that did not contain subway-only trips were discarded. Eventually, 370,033 trips were selected. The study was based on OD information collected in 2012.

A sample traveler was presented to describe the travel plan shown in Figure 8 and Figure 9. The associated OD distribution and the related subway lines and roads are shown in Figure 8. Shixia Station and Huaxin Station are the two transfer stations for Lines 3 and 7 in the subway system. Figure 9 indicates the possible travel routes. Specifically, segments A and F represent walking trips on the road network from home to the originating station of the subway trip and from the terminal station of the subway trip to the workplace. For this traveler, Shangsha Station on Line 7 is the originating station, and Shaibu Station on Line 3 is the terminal station. Therefore, a transfer from Line 7 to Line 3 at one of the transfer stations above is necessary. Segments B and E are the parts of the subway trip that are shared between the two transfer schemes. Segment C refers to the part of the subway trip that involves transferring at Shixia Station, and segment D is selected when transferring at Huaxin Station. In general, both schemes are provided by the BMS and sorted using evaluation indexes, such as time and financial cost. The less time-consuming way is adopted in this study, and Shixia Station is chosen as the transfer station because segment D is more time-consuming, due to its longer length and the greater number of stations it passes than segment C. Therefore, the detailed travel plan for sample traveler is composed of segments A, B, C, E and F and can be written out as follows:Step 1. Depart and walk 772 m from home to Shangsha Station.Step 2. At Shangsha Station, take Line 7 to Shixia Station.Step 3. Transfer, walking 94 m.Step 4. At Shixia Station, take Line 3 to Shaibu Station.Step 5. Walk 644 m from Shaibu Station to arrive at the workplace.

The empirically determined travel time returned by BMS is 53 min 15 s. All of the travelers involved in the experiment have their generated trips in the same way, and the trips are utilized in dynamic traffic simulations in the next section.

### 3.3. Travel Process Simulation

An agent-based traffic simulation framework was designed and developed to simulate the travel process of the chosen 370,033 travelers taking the trips described above. The proposed framework aimed to simulate the behaviors of all of the traffic entities chronologically. Because the situation examined here involves urban subway commuters going to work during the morning peak hours, the essential restraint was not being late. Thus, all of the travelers were required to arrive at their workplaces before a specific time (9:00 a.m.).

Multiple iterations were executed before the system reached a relaxed state. The departure time for each traveler was calculated using the empirically determined travel time returned by BMS in the first iteration. For example, when the empirically determined travel time was 1 h, then the departure time should have been 8:00 a.m. to avoid being late. The departure time needed to be adjusted when a late arrival occurred.

$T_{i+1}$ and $T_{i}$ in Formula (7) represent the departure times for two adjacent iterations, and δ indicates the adjusted timespan. δ is assigned to be −1; thus, when a person was late, his or her departure time was changed to 1 min earlier than in the former iteration.
(7)$$T_{i+1}=T_{i}+\delta$$

In this experiment, 116 iterations were carried out before the system reached a relaxed state. Traffic characteristics, such as times and the events performed, were output and analyzed to extract passenger flow characteristics and estimate congestion using the calculation methods discussed in Section 2.3.

## 4. Results and Discussion

### 4.1. Predicted Congestion Distribution

The passenger flow distribution generated from the traffic simulation results reflected the passenger flow volume in subway segments during morning peak hours, and the load factors of the subway vehicles could be calculated using Formula (5), which indicates the degree of congestion for every segment. The number of passengers selected for evaluation in the experiment was nearly 40% of the real number of daily subway passenger-trips made during the morning peak hours; therefore, μ was assigned to be 0.4. In addition, our simulation needed to fully include the peak hours in our simulation period. It takes a certain amount of time for the traveler to travel from the departure to the stop and from the stop to the work place. So according to empirical data, the simulation extended from 6:30 a.m. to 9:00 a.m.

The parameters related to the Shenzhen subway operation are shown in Table 3. According to the reference data, the congestion situations were divided into three levels, smooth, crowded and congested. Smooth means that there was no congestion, and the subway segment operated unimpeded. The crowded and congested conditions both refer to large numbers of people in the carriages, and congested describes a worse situation. Figure 10 indicates the vehicles’ load factors, which represent the predicted degree of congestion of the subway segments. The classification thresholds in Figure 10 were calculated according to the reference data, which provided the ranges of passenger flow volumes for the three congestion levels. The corresponding ranges of load factors of the vehicles for these three congestion levels were calculated using the same calculation method, and the load factor thresholds were 0.45 and 0.70.

### 4.2. Reference Congestion Distribution

Statistics on the congestion conditions of the subway segments have been extracted from the daily morning peak hours’ congestion information released by the Shenzhen metro operation microblog (see Figure A1 in Appendix A) in order to verify the accuracy of the predictions. Congestion information was collected for the 15 days between 20 April and 12 May, excluding weekends and holidays. The statistical results are presented in Table A1, and D1, D2, and D3 represent the congested, crowded and smooth days, respectively. According to an enquiry to their staff, the passenger flow can be divided into three levels. Specifically, the green line represents 10,000–20,000 passengers per hour (C1), the yellow line represents 20,000–30,000 per hour (C2) and the red line represents over 30,000 passengers per hour (C3). In this experiment, the values of C1, C2, and C3 were set to 15,000, 25,000, and 30,000, respectively. The formula for the reference load factor can be written as follows:(8)$$\beta =\frac{\sum_{i=0}^{3}C_{i}D_{i}}{d\;\times {\;C}_{p}\times {\;N}_{1}}$$

Here, d represents the number of days for which congestion information was collected; this value is set to 15 here. $C_{p}$ represents the designed passenger capacity of the subway vehicles; ${\;N}_{1}$ indicates the number of trains that depart per hour. Figure 11 indicates the load factors of the vehicles calculated from the statistical congestion information.

### 4.3. Performance Assessment

According to the results in Table 4, the matching ratios for Lines 3 and 11 are encouraging. Figure 12 and Figure 13 present a comparison of the reference data and predictions for Lines 3 and 11.

Figure 12 shows three consistent segments:Yitian Station to Futian Station (three stations) is labeled as smooth in both the reference data and the predicted results.Laojie Station to Mumianwan Station (seven stations) is labeled as congested in both the reference data and the predicted results.Danzhutou Station to Shuanglong Station (11 stations) is labeled as smooth in both the reference data and the predicted results.

There are also two inconsistent segments:Futian Station to Laojie Station (six stations) is labeled as smooth in the reference data and crowded in the predicted results.Mumianwan Station to Danzhutou Station (two stations) is labeled as congested in the reference data and crowded in the predicted results.

According to the reference data, the segments from Laojie Station to Danzhutou Station on Line 3 are frequently congested, and the proposed method successfully detects this behavior. Figure 12 shows that the congestion on Line 3 between Laojie Station and Danzhutou Station and including other segments is unhindered. In addition, the predictions overestimate the degree of congestion on inconsistent segment (1) and underestimate the degree of congestion on inconsistent segment (2).

Figure 13 shows three consistent segments:(1)Bitou Station to Ma’an Hill Station (four stations) is labeled as smooth in both the reference data and the predicted results.(2)Fuyong Station to Houhai Station (seven stations) is labeled as crowded in both the reference data and the predicted results.(3)Houhai Station to Futian Station (three stations) is labeled as smooth in both the reference data and the predicted results.

There is also one inconsistent segment:

Ma’an Hill Station to Fuyong Station (three stations) is labeled as smooth in the reference data and crowded in the predicted results.

The predicted results for Line 11 are promising. The major crowded segments are well detected, and only three segments are incorrectly assessed.

The values of the matching ratio are relatively low for Lines 4 and 7. Figure 14 and Figure 15 compare the reference and predicted congestion for these lines.

The comparison shown in Figure 14 indicates that the predicted results do not reflect the recurrent crowded situation on Line 4. However, this seemingly unsatisfactory result can be clearly explained by taking into account the assumptions of the experiment and the actual conditions of the subway system of Shenzhen. The possible reasons are as follows:(1)As noted above, the prediction of congestion in the subway system of Shenzhen is based on the simulation of subway commuting behaviors. One significant assumption is that the commuting passenger flow makes up a large proportion of the total flow, and non-commuting passenger flow can be neglected. This assumption is reasonable for the subway lines other than Line 4, which includes Shenzhen North subway station that connects to Shenzhen North railway station (SZNRS). SZNRS serves a huge number of passengers traveling in and out of Shenzhen. Therefore, Shenzhen North subway station is constantly over capacity; its year-round passenger volume reached 67 million in 2015, ranking first among all of the stations. In contrast, in this experiment, the passenger flow volume for the Shenzhen North Station is 1027 and ranks 117th. Line 4 is the major subway line that connects Futian District (the downtown area) with SZNRS; thus, it carries a very large non-commuting passenger flow. This study does not consider the influence of non-commuting passenger flows, and the overall congestion conditions are underestimated for Line 4.(2)The OD information used in this study was collected in 2012. The LongHua New District has developed rapidly in recent years, and many people have moved there. Shenzhen Line 4 is the only subway line that reaches LongHua New District, and this line connects the LongHua New District with downtown. Thus, the demand for travel on Line 4 has increased rapidly in recent years, which may have produced this inconsistency.(3)Line 4 is the only one operated by MTR <Shenzhen>; the others are operated by SZMC. Different operation and management practices may have some influence on the passenger flow characteristics.

The comparison shown in Figure 15 indicates that the predicted result and the reference data are consistent for the western portion of Line 7, namely the segments from Xili Lake Station to Chegongmiao Station. In addition, almost all of the other segments are inconsistent (except for the segment between Shangsha Station and Shawei Station). The reason for this phenomenon may be that Chegongmiao Station is the only transfer station that serves four lines and is the most significant transportation junction in the subway system of Shenzhen. In this study, only those citizens traveling by subway are considered. The results confirm the importance of Chegongmiao Station has been confirmed. The passenger flow volume of Chegongmiao Station is 27, 122, ranking first. Moreover, a great number of passengers who transfer at Chegongmiao Station get to Line 7, increasing the passenger flow volume of the segments after Chegongmiao Station on Line 7. Thus, the predicted results include overestimates for these segments.

## 5. Conclusions

The prediction of subway congestion is a significant issue, and such predictions can be used to guide the travel of residents and provide decision support for traffic managers and transit operators. In this study, a hybrid method that combines a static traffic assignment model with an agent-based dynamic traffic simulation model is proposed and tested in a case study city. The proposed method includes two major stages. For the traffic assignment stage, the trips of the subway commuters are generated from the OD information by calling BMS. During the prediction stage, an agent-based framework is developed, and an agent-based traffic simulation model is applied to simulate the travel process for all of the trips mentioned above. The predictions are then compared with the reference data. The experimental results show that
(1)promising results are obtained by the proposed method;(2)data on the workplaces and homes of residents can be utilized to forecast passenger flows in public transportation systems; and(3)the predicted congested subway segments and the corresponding causes can be provided for use in traffic management-related decision support.

Traffic congestion, especially during peak hours, has become a challenge for transportation systems in many metropolitan areas. Many studies have examined the prediction of congestion; however, these studies focus mainly on road traffic, and subway transit has seldom been examined. This study provides a feasible method to predict subway traffic congestion during peak hours. Compared with the actual congestion situation of the subway during the morning peak hours, the results of our study are encouraging.

The forecasting of traffic flow has always been an important research content of sensor network. Dynamic traffic simulation models are often used to predict traffic flow. However, the method is hard to do in subway traffic prediction due to the lack of information on the origins and destinations of passengers. A hybrid method that combines a static traffic assignment model with an agent-based dynamic traffic simulation model proposed in this study can make it feasible. The method of constructing an OD matrix from commuters’ home addresses and work places and assigning them to the subway transportation network makes up for the shortcomings in the data. It provides a new idea in the case of using OD information which is relatively easy-to-obtain data to predict subway traffic flow. This study could be taken as an example for similar studies that model subway traffic in other cities.

In addition, the actual simulation scenario in this study is slightly idealized. The flow of non-commuting passenger during peak hours is not taken into account. In the future, unexpected factors, such as accidents and the effects of weather; passengers stop or delay in situations of actual congestion, will be taken into consideration to permit improvement of the proposed method. This study only considers commuting by subway. Prediction of the occurrence of congestion in multimodal traffic, including subways, buses, and taxis, will also be explored in our future work.

In addition, the proposed approach has a drawback, in that the flow of non-commuting passengers during peak hours is not taken into consideration.

Compared with the actual congestion situation of the subway during the morning peak hours, the results of our study are encouraging. The inconsistent lines have also been reasonably explained based on the actual situation. The research results show that it’s reasonable and effective for predicting congestion in a subway system during peak hours by the hybrid method to estimate recurrent congestion in this subway system. This study provides a feasible method to predict subway congestion. It can provide a new idea in the case of using relatively easy-to-obtain data such as OD information.

In addition, the actual simulation scenario in this study is slightly idealized. In the future, unexpected factors, such as accidents and the effects of weather; passengers stop or delay in situations of actual congestion, will be taken into consideration to permit improvement of the proposed method. This study only considers commuting by subway. Prediction of the occurrence of congestion in multimodal traffic, including subways, buses, and taxis, will also be explored in our future work.

## Figures and Tables

**Figure 1 sensors-20-00150-f001:**
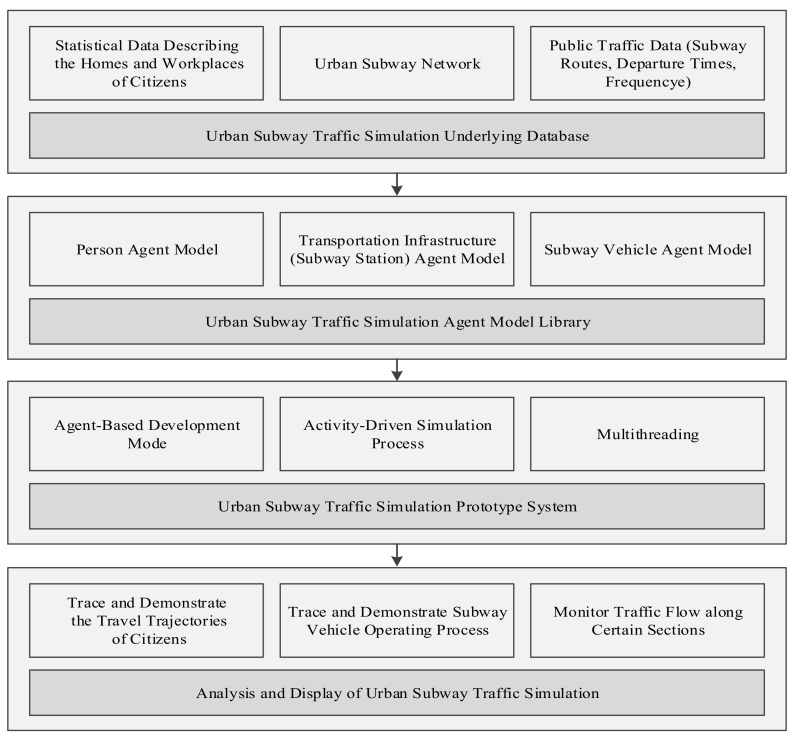
Traffic simulation framework architecture diagram.

**Figure 2 sensors-20-00150-f002:**
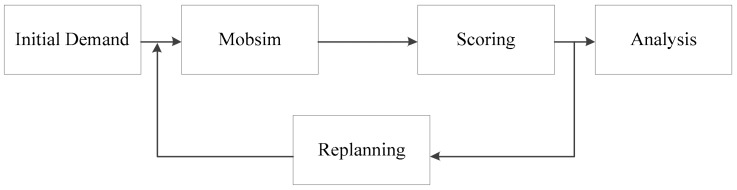
Simulation workflow.

**Figure 3 sensors-20-00150-f003:**
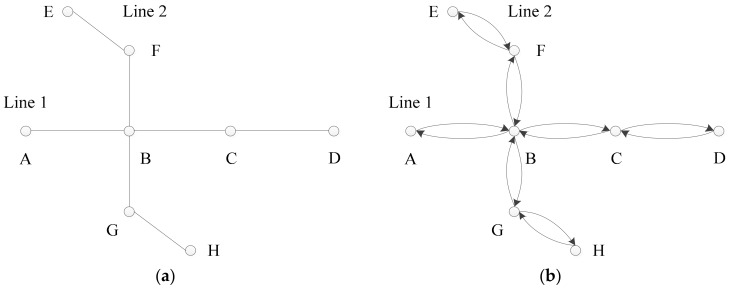
(**a**) A sample subway network with eight nodes and seven edges. (**b**) Passenger flow network with eight nodes and 14 directed edges.

**Figure 4 sensors-20-00150-f004:**
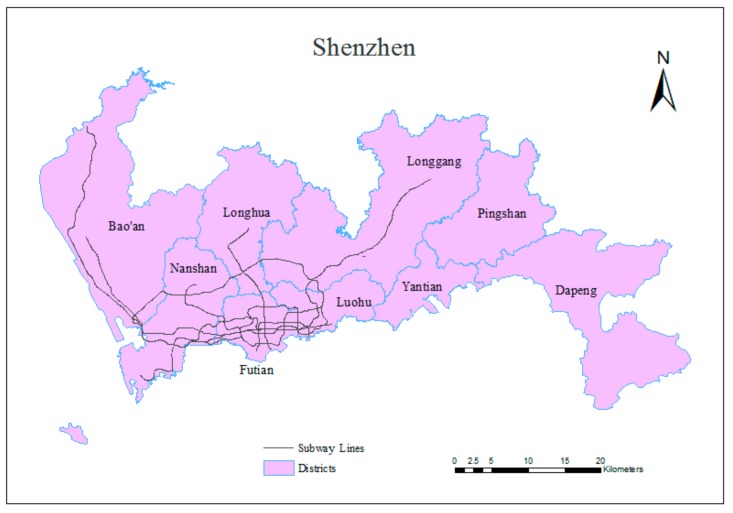
Study area.

**Figure 5 sensors-20-00150-f005:**
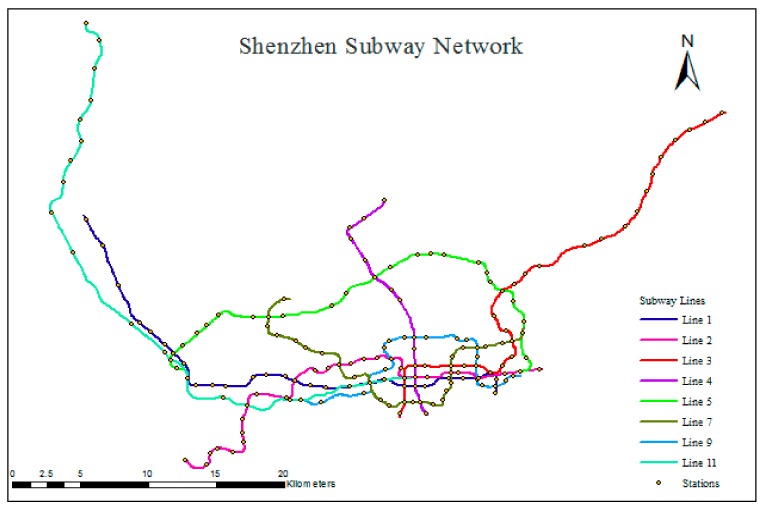
Diagram showing the structure of the subway network of Shenzhen.

**Figure 6 sensors-20-00150-f006:**
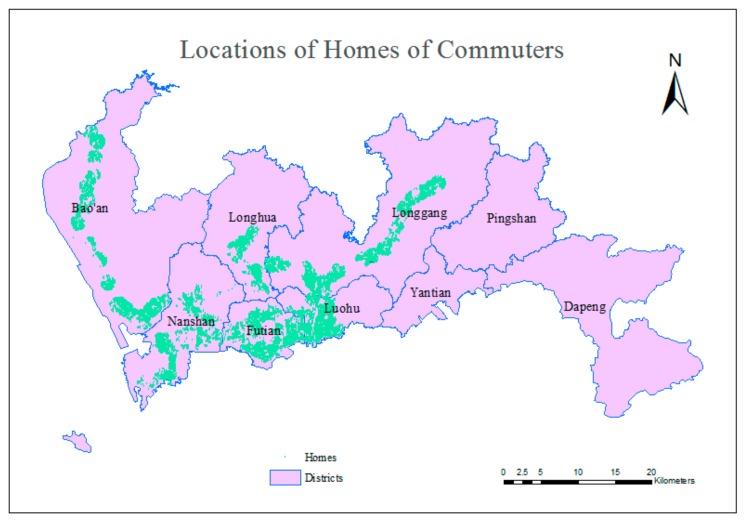
Locations of the homes of selected citizens.

**Figure 7 sensors-20-00150-f007:**
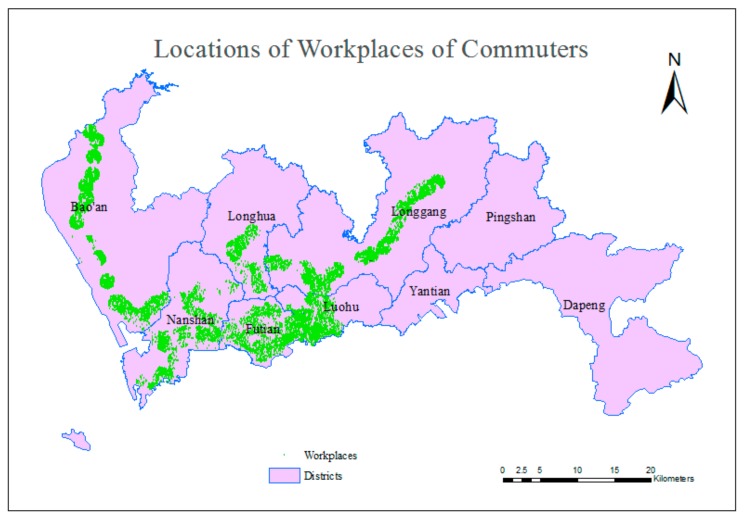
Locations of the workplaces of selected citizens.

**Figure 8 sensors-20-00150-f008:**
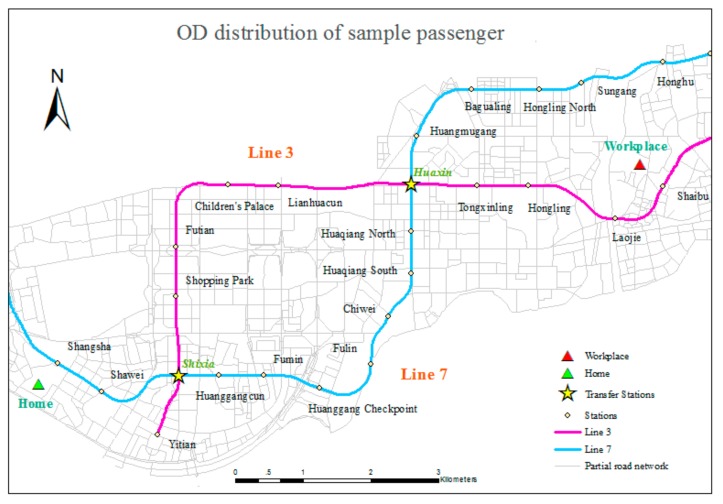
OD distribution of a sample passenger.

**Figure 9 sensors-20-00150-f009:**
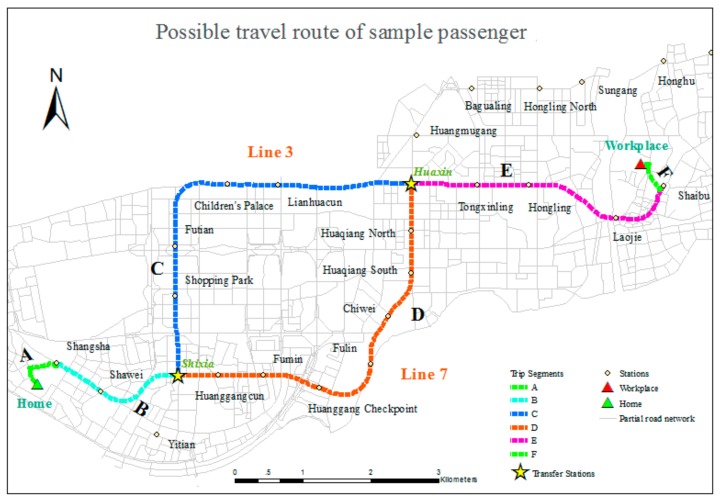
Possible travel routes of a sample passenger.

**Figure 10 sensors-20-00150-f010:**
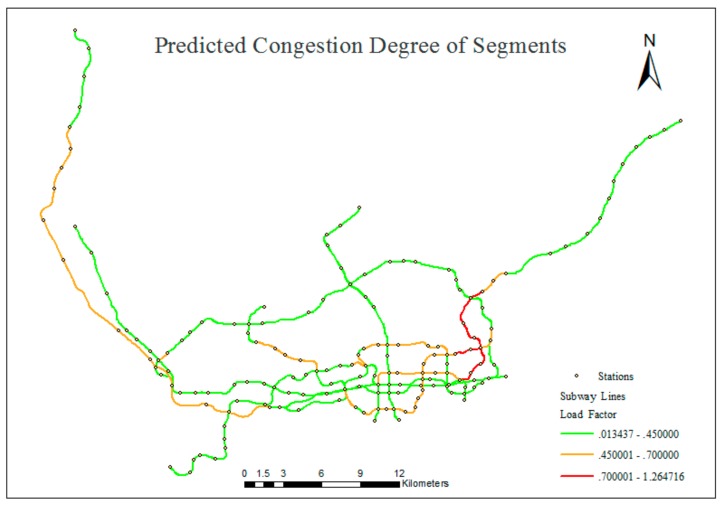
Predicted degree of congestion of subway segments.

**Figure 11 sensors-20-00150-f011:**
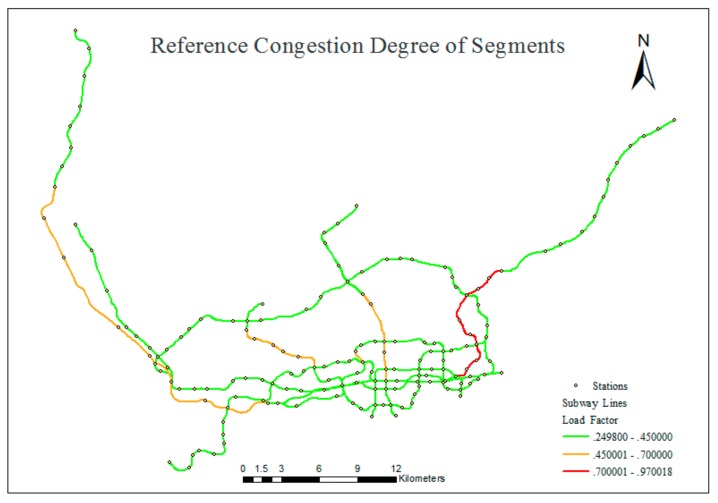
Reference degree of congestion of the subway segments.

**Figure 12 sensors-20-00150-f012:**
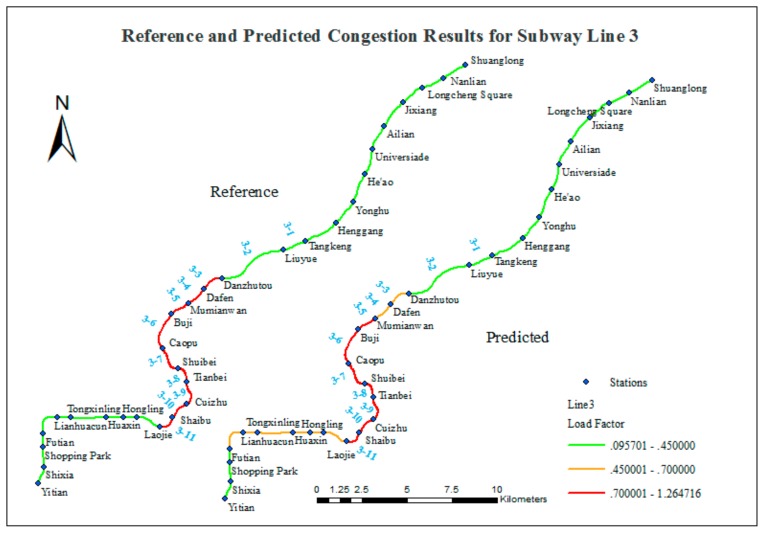
Comparison of reference and predicted congestion results for Line 3.

**Figure 13 sensors-20-00150-f013:**
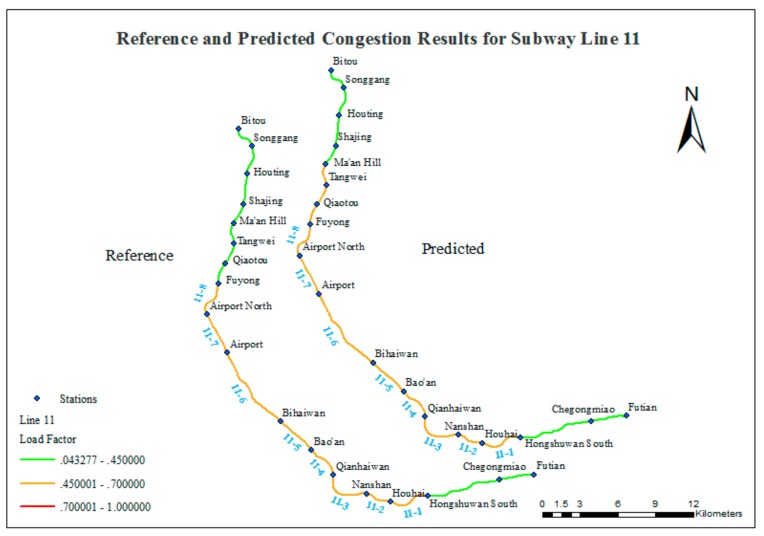
Comparison of reference and predicted congestion results for Line 11.

**Figure 14 sensors-20-00150-f014:**
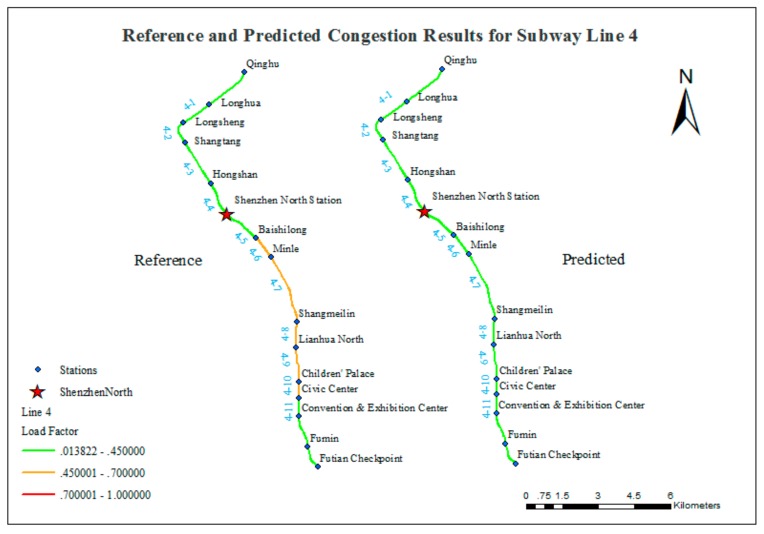
Comparison of reference and predicted congestion results for Line 4.

**Figure 15 sensors-20-00150-f015:**
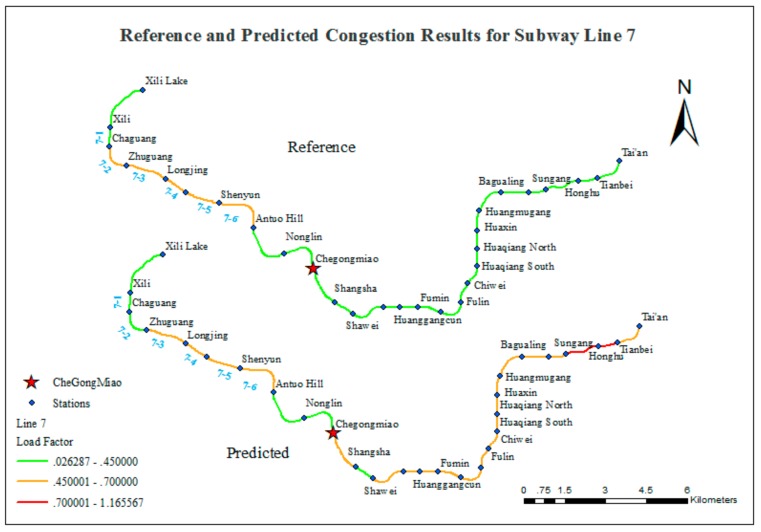
Comparison of reference and predicted congestion results for Line 7.

**Table 1 sensors-20-00150-t001:** Passenger flow volume and annual change for major modes of transportation.

	2013	2014	2015	2016	2017
**Bus**	2201.78 (−3.56%)	2257.39 (2.53%)	2068.92 (−8.35%)	1867.99 (−9.71%)	1654.25 (−11.44%)
**Taxi**	432.30 (+5.99%)	438.42 (1.42%)	391.13 (−10.79%)	373.62 (−4.48%)	370.8 (−0.76%)
**Subway**	917.15 (+17.39%)	1036.75 (+13.04%)	1121.88 (+8.21%)	1297.13 (+15.62%)	1655.45 (+27.62%)

Passenger flow volume unit: million trips.

**Table 2 sensors-20-00150-t002:** Information on the subway lines of Shenzhen.

Subway Route	Originating Station	Terminal Station	Stations	Length (km)
Line 1	Luohu	Airport East	30	41.0
Line 2	Chiwan	Xinxiu	29	35.7
Line 3	Suanglong	Yitian	30	41.7
Line 4	Futian Checkpoint	Qinghu	15	20.5
Line 5	Qianhaiwan	Huangbeiling	27	40.0
Line 7	Xili Lake	Tai’an	28	30.1
Line 9	Hongshuwan South	Wenjin	22	25.4
Line 11	Futian	Bitou	18	51.9

**Table 3 sensors-20-00150-t003:** Operation parameters of the Shenzhen subway system.

Subway Line	Peak Interval (7:30–9:00)	Common Interval	$N_{t}$	Marshalling	$C_{p}$
Line 1	2 min	4 min	60	6A	2502
Line 2	3.5 min	5 min	38	6A	2502
Line 3	3 min	5 min	42	6B	1800
Line 4	2.5 min	5 min	48	6A	2502
Line 5	3.5 min	5 min	38	6A	2502
Line 7	5 min	8 min	26	6A	2502
Line 9	5 min	8 min	26	6A	2502
Line 11	5 min	8.5 min	26	8A	2564

**Table 4 sensors-20-00150-t004:** Comparison between actual and predicted results.

Subway Route	Reference Segments			Predicted Segments			Number of Matches	Matching Ratio
	Congested	Crowded	Smooth	Congested	Crowded	Smooth		
Line 1	0	0	29	0	0	29	29/29	1.000
Line 2	0	0	28	0	1	27	27/28	0.964
Line 3	9	0	20	7	8	14	21/29	0.724
Line 4	0	5	9	0	0	14	9/14	0.642
Line 5	0	0	26	0	1	25	25/26	0.962
Line 7	0	5	22	3	18	6	9/27	0.333
Line 9	0	1	20	1	6	14	15/21	0.714
Line 11	0	8	9	0	6	11	14/17	0.824
Total	9	19	163	11	40	140	149/191	0.780

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
