# Peer review of "A Hybrid Method for Predicting Traffic Congestion during Peak Hours in the Subway System of Shenzhen"

_sensors, 2019, doi:10.3390/s20010150_

Round 1
Reviewer 1 Report
The authors propose a method based on static traffic assignment and simulation to predict the congestion levels on links of a subway network. The authors then put forth a case study with real data from the Chinese City of Shenzhen, in which they compare the outputs of the model with data on congestion levels. My main concern regarding this paper is that I see no contribution to the theory or the practice in the method proposed by the authors.
Using traffic assignment and simulation models to predict traffic congestion in subway is a well known application in the literature. The sole proposal of a model and comparison of its outputs with real data is not sufficient contribution in this case, since other models already exist. The authors should address which improvement their model offers relative to other models already available. This is not clear in the paper.
Reviewer 2 Report
A novel method is proposed in this paper to estimate the demand and metro traffic flow during peak hours. Based on the OD matrix extracted from social insurance data, travel path of each person was derived using BMS. After filtering the travelers using solely subway for commuting, an agent-based simulation method was utilized to get the congestion status of the metro segments. The methodology of this study is sound.
Minor Concerns:
There are many careless mistakes in the manuscript. On page 1, line 27, "… used as are ference …" should be "... used as a reference …". On page 2, line 45, "… have used arious …" should be "… have used various …". On page 7, Table 1, "Subwa" should be "Subway". On pate 10, line 319, "indynamic …" should be "in dynamic …". The penetration rate of the social insurance data could be provided if possible. Queuing sometimes happens in severely congested stations. How does the agent-based simulation system model this phenomenon? On page 3, it is mentioned that the popularization of map apps ensure the consistency of the generated trips with reality. However, most commuters choose the travel path based on their experience instead of apps. The generated trips are valid may rely on the accuracy of the recommended path of the map apps.Major Concerns:
Is the spatial buffer analysis conducted simply based on radius or real road network? The distance may be longer than direct distance due to the detour of road network. Section 3.3, it is mentioned that the commuters will departure earlier exactly based on the empirical travel time. However, in real life, many people tend to reserve some time considering possible delay, or going for a breakfast. How do you consider this problem and how will this affect the results? Not all people who live near metro stations tend to commute by subway, while this study seems to consider that they will take subway between home and work place other than using other transportation mode. Please clarify the rationality of this assumption. How is the congestion status released by Shenzhen operation group estimated? Please explain.Author Response
Please see the attachment.

Reviewer 3 Report
line 27 – “as are ference” – from the context of the sentence it should be replaced into “as a reference”
line 71 – “It based on” – should be changed into “it is based on” or “based on”
line 158 – Fig. 1 has no “introduction” in the previous section. It should be referred to more than once; there is a mistake in the word “frequency”
line 191 – “relaxed state is defined as the first time when none of the travelers are late” – what do you mean when you say that somebody is late? What is the minimum unit of delay? Is it measured in seconds or minutes? … and ‘travellers’ is an appropriate form
line 231: “factorof „ -> factor of
line 215 - 240 – there is a problem with readability – spaces between symbols and explanations are not created in a proper way – need correction
Section 3.1 - it seems that the authors, generally, in this section neglect to cite and refer to sources of data taken into case study. It is explained in details below:
Figure 4 - What was the source for subway lines and districts? Was fig. 4 made by authors or was obtained from another source?
line 252 “Table 1 records the passenger flow volume and annual changes for these three modes of transportation” – on the basis of what? Was it published anywhere before? It seems like the authors have counted the flows themselves. Is it the right assumption?
Figure 5 - What was the source for subway lines and districts? Was fig. 4 made by authors or was obtained from another source?
line 260 – 272 – what is the source for this kind of information?
line 275 – Table 2 - Was it published anywhere before? It seems like the authors have counted the flows themselves.
line 285- 286 “morning peak hours on workdays” – how did you choose or estimated ‘morning peak hours’? What is the hourly interval? Is it similar to this observed in other metropolitan cities in China and/or in other large and densely populated cities?
Figure 6 and 7 – Where did you get the data from for the ‘buffering’ presented on figures?
line 350 “In addition, the simulation extends from 6:30 am to 9:00 am”. – this time span has not been justified either earlier or in this part of article. It should be supplemented.
line 392 – The headings in Table 4 are hard to read or understand – this should be changed.
line 448 – “The OD information used in this study was collected in 2012”. – this sentence appears much too late. Authors should specify the time period of data collection in section 2 or 3.
Conclusions – this section is to short.
The authors should answer the following questions:
- What from the above research can be implemented in other cities in the world taking into account the availability of data and the subway specification?
- What does the analyses depend on the most and what are they vulnerable to?
- As the authors wrote, the data collection for the OD was made in 2012 and now is 2019. Has anything changed since then? What are the general tendencies in the case study city?
– Have you thought about possible stops during the travel or delays which travellers have no influence on? How can you modify your methodology in future in order to take them into consideration?
Round 2
Reviewer 1 Report
I am still not convinced of the novelty of this research. Applying static traffic assignment or a simulation model to predict link congestion in a subway network is nowadays a fairly standard application. Many technical studies on subways around the world already use this methodology embedded in commercial software, such as TransCAD, AIMSUN and VISSIM. In the past, these studies were hard to do due to the lack of information on the origins and destinations of passengers. This was the main hurdle. However, with the advent of smart card technology, we can gather large amounts of data on origin and destination of passengers in subway systems, so that the main problem, the estimation of the OD matrix, is practically solved. The main research question now is what competitor models best fits to the data. For example, there are many alternative traffic assignment models: DUE, SUE, Logit SUE, Probit SUE etc. Which one best fits to subways? This is a relevant question which could be addressed in your research. I still think the paper is more relevant to a technical audience, and not to researchers in the field.
Reviewer 2 Report
All suggestions and comments from the first-round review have been addressed.
Reviewer 3 Report
Please, see the attachment.
